# A Five-Step Approach to Planning Data-Driven Digital Twins for Discrete Manufacturing Systems

**Matevz Resman, Jernej Protner, Marko Simic *** and **Niko Herakovic**

Faculty of Mechanical Engineering, University of Ljubljana, 1000 Ljubljana, Slovenia;
matevz.resman@fs.uni-lj.si (M.R.); jernej.protner@fs.uni-lj.si (J.P.); niko.herakovic@fs.uni-lj.si (N.H.)
* Correspondence: marko.simic@fs.uni-lj.si; Tel.: +386-1-4771-727

**Abstract:** A digital twin of a manufacturing system is a digital copy of the physical manufacturing system that consists of various digital models at multiple scales and levels. Digital twins that communicate with their physical counterparts throughout their lifecycle are the basis for data-driven factories. The problem with developing digital models that form the digital twin is that they operate with large amounts of heterogeneous data. Since the models represent simplifications of the physical world, managing the heterogeneous data and linking the data with the digital twin represent a challenge. The paper proposes a five-step approach to planning data-driven digital twins of manufacturing systems and their processes. The approach guides the user from breaking down the system and the underlying building blocks of the processes into four groups. The development of a digital model includes predefined necessary parameters that allow a digital model connecting with a real manufacturing system. The connection enables the control of the real manufacturing system and allows the creation of the digital twin. Presentation and visualization of a system functioning based on the digital twin for different participants is presented in the last step. The suitability of the approach for the industrial environment is illustrated using the case study of planning the digital twin for material logistics of the manufacturing system.

**Keywords:** data-driven factory; digital model; digital twin; modelling; discrete-event simulation





## 1. Introduction

All manufacturing systems, and thus also a discrete manufacturing system (where distinct items are manufactured [1]) should be more agile, flexible, and sustainable to cope with the dynamic changes in the manufacturing environment and market demands, while maintaining the quality of manufactured products [2–4]. Recently, this has been addressed by the concept of data-driven factories, which are characterized by the following features: agility, learning capability, and human-oriented manufacturing [5]. Often referred to as smart-manufacturing systems, these systems aim to convert the data collected throughout the entire manufacturing process into manufacturing intelligence. In turn, manufacturing intelligence enables knowledge-driven decisions that positively impact the operation of the entire manufacturing system [6,7]. The data-driven factory represents a leap forward from the more traditional automation to a fully connected and flexible system [8,9]. To develop the data-driven digital factory, digital models are a key enabling technology.

A digital model is a simplified representation of a real system and process, with the characteristics which reflect the properties and explain the behavior of the real manufacturing system. It represents a partial description of the real system, because it is made for a specific purpose and therefore cannot include all the details of the real system. Figure 1 shows the connections between a real manufacturing system and its digital model. To synchronize both, data exchange should be performed in both directions.

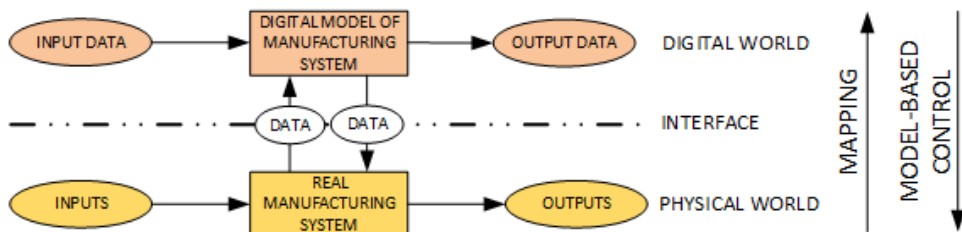

**Figure 1.** Connections between a real manufacturing system and its digital model.

Digital models must keep up with the changes in the real system, and change accordingly over time, for which it is essential to structure the data flow from the real environment to the model and vice versa. When a digital model and a real system are bi-directionally connected by a feedback control loop, the model is usually referred to as a digital twin, which was first mentioned in 2003 [10]. However, the concept of a digital twin was envisioned well before the technology necessary for its implementation became available. Various definitions of a digital twin can be found in literature [11,12], one of which was proposed by NASA in 2010. It defines the digital twin as an ultra-realistic, high-scaling simulation model that uses the best-available physical models, with data being collected throughout the whole lifecycle of the system [13,14].

For the purposes of our research, we used the following definitions [15]. The term *digital model* is used for a model where there is no automated data exchange between the physical world and the digital world, and in which all the data exchange is conducted manually. The second term, *digital shadow*, is used for an automated, one-way data flow from the physical world to the digital world, but not vice versa. The *digital twin* is used when a physical object and its digital world are connected in both directions to perform an automated two-way data flow.

Data flows are essential in the development of digital twins. By creating a link between the real system and its digital twin, a data-driven digital twin is created; a digital twin that evolves along with its physical counterpart based on the actual data acquired from the real system.

In manufacturing, there are large amounts of structured, semi-structured and unstructured data generated during a product's lifecycle. The data is collected automatically in real time and represents the basis for the development of the digital twin [16,17]. The problem with planning and the development of a digital twin is that there is a variety of different data sources, types, and structures. For example, data sources include data from sensors on machines, billboards, work plans, operating schedules, data from a Measurement Execution System (MES), Enterprise Resource Planning (ERP), etc. These data take various forms such as graphs, tabular records, collected data from sensors and they are on different scales in terms of both time and volume. Furthermore, there is no consensus on what parameters and data are necessary for planning a data-driven digital twin. The definition of parameters necessary for the development of the digital twin in manufacturing systems is a topic of interest to many researchers. However, a general approach to the definition of the necessary parameters is still deficient. This paper addresses exactly this problem, i.e., it presents a five-step approach which enables the assessment of which data needs to be captured and in which form it should be obtained.

One of the most important requirements of modern manufacturing systems is the ability to perform real-time process optimization and control. To achieve this goal, it is necessary to design suitable digital models, perform "what-if" simulations and establish a feedback control loop with the physical system in order to develop a digital twin. For this reason, various manufacturing-systems frameworks, theories, and models have been discussed in [18]. Discrete-event digital models are most commonly used to simulate discrete manufacturing systems [19–21]. This is the reason why we used such models for the development of the digital twin in our approach. However, these papers do not show the exact procedure or steps for developing the digital twin, but only how

to use it. How to start developing a digital model is described in the paper [22]. A methodological approach for the integration of a virtual engineering-environment platform with a simulation tool is presented in [23,24]. The presented methodology includes the gathering of manufacturing data from the shop floor. With the help of a mathematical model, the material flow in the manufacturing system and the use of a simulation tool for visualization are described, but they do not focus on the development of the digital twin and connection of the digital model with the real manufacturing system. Researchers [25–28] have focused on communication and data exchange between assets in a manufacturing system throughout the entire manufacturing life cycle and the development of a framework to monitor the current status in realtime, track past events, and support decision making by predicting the future events.

As already mentioned, various interpretations of digital twins are used by researchers [29,30]. The first view presents the digital twin as the kind of model from which different types of simulations can be derived. The second view defines the digital twin as a simulation. The sum of different simulation models and systems is what makes up the digital twin useful. Another interpretation of the digital twin was used by the authors Liu et al. [31], who used the digital twin to configure, control, and optimize a smart manufacturing system. We use a very similar interpretation of the digital twin in our approach, but our research also focuses on the development of the digital twin, not just on the presentation of advantages of using the digital twin in manufacturing.

Authors Stark et al. [32] used the digital twin as a testing method in a new approach to the next-generation manufacturing system. By applying this methodology, users are able to develop, validate, and optimize the system with a digital model, but the methodology does not include the feedback loop with the real manufacturing system as proposed in our approach. The authors [12,33,34] introduced digital twin shop-floor to present the interaction and convergence between physical and virtual spaces. Data from both spaces as well as the merged data are provided to drive the production. They also propose a new methodology for the digital twin-driven product design, manufacturing, and service to solve the problems about data in product lifecycle. Their research mainly focuses on data management rather than how to develop the digital twin in software tools for logistics and how to control the real manufacturing system with the digital twin, as described in our proposed approach. The authors Zhuang et al. [35] developed a framework for smart production management and production control based on the digital twin for an assembly shop-floor. Their work does not include steps for the user to develop a digital twin of the manufacturing system from the initial idea to the final implementation.

To the best of our knowledge, no research exists that considers the multi-step approach with the instructions to users for the development of the digital twin and the visualization of the output data as a whole (from the initial idea to the final presentation of the results). Various research groups [19–24,31–35] focus on different segments of the digital twin planning process, but none of them suggest a comprehensive approach to planning and development of a data-driven digital twin. Hence, our paper defines the necessary data for modelling the digital model and presents the strategy to connect the digital model to a real manufacturing system to create a digital twin. The paper also addresses the presentation of the manufacturing system output data that improves the working environment for various users of the digital twin, in terms of easier planning and keeping up with the manufacturing schedule, detection of bottlenecks, etc. Therefore, we have proposed an approach that guides the user step-by-step from the development of the digital model, to the connection between the digital model and the real world through a feedback loop, to the control of a real system by the digital twin. Finally, the presentation of the output data for various participants is given. The approach is illustrated and validated in a case study of internal logistics (i.e., material flow) of a laboratory-scale discrete manufacturing system.

Our study has several contributions regarding to the given literature. First, we proposed the classification of the building blocks of manufacturing systems into four groups, regardless of the type and size of the manufacturing system. Second, we develop

and present a new approach for defining the parameters needed for planning the digital model for each building block. Third, we describe the innovative control of the real manufacturing system with its digital twin. For this purpose, the real manufacturing system is connected to its digital model bi-directionally, which defines the parameters that need to be exchanged. The overall contribution combines all partial contributions and results mentioned above into a new five-step approach, allowing to develop a digital twin of manufacturing system in a more streamlined way.

The rest of this paper is organized as follows. The Materials and Methods section presents our five-step approach to developing the digital twin. Each step is described in detail in its subsection. In the first step, users get familiarized with the manufacturing system; in the second step with the sequence of the processes; in the third step with the parameters required to develop a digital model of the manufacturing system; in the fourth step with the connection of the digital model to the real manufacturing system and finally, in the fifth step with the benefits of using a digital twin. To illustrate the approach, a case study is presented in Section 3. The case study presents the real manufacturing system with its digital twin developed using our five-step approach. We demonstrate the manufacturing system supported with the digital twin, which has been tested and validated in terms of functionality and practical applicability. The Results and Discussion section shows all the steps with values and graphical representations for the case study. Finally, Conclusions highlights the main findings as well as future work.

## 2. Materials and Methods

Modern manufacturing systems and processes are complex due to increasing number of product variants and are therefore difficult to bring into the digital world, but complexity is necessary to stay competitive [36]. They must be developed with sustainable organization in mind. In this way, we can reduce the consumption of raw materials and improve working conditions as well as perform better management of resources and operational activities along the assembly line [37,38]. One of the possibilities for increasing the competitiveness is the development of digital twins capable of performing what-if scenarios. From our point of view, what-if scenarios are different simulation runs in a simulation software tool (e.g., Siemens Plant Simulation, Visual Components, Delmia). The results of what-if scenarios show the matrix of output parameters according to the different input parameters. Competitiveness is also the reason to introduce the approach for developing data-driven digital twins for discrete manufacturing systems.

To develop the digital twin as a real-time representation, the physical object needs to be digitized [39–41], and appropriate parameters (engineering and operating) need to be defined. The engineering parameters describe the properties of the manufacturing system (e.g., dimensions, velocity, and engine power), while the operating parameters describe the properties of the corresponding manufacturing processes. The proposed approach describes the input parameters needed to develop a digital twin of the desired manufacturing system. Based on the input parameters and the constraints of the manufacturing system, the result of implementing the digital twin is to control the real manufacturing system in real time, for which algorithms are used to determine the sub-optimal manufacturing plan.

The five-step approach (each step is highlighted in Figure 2) proposes a step-by-step development of a digital twin, regardless of whether the input data is gathered automatically or manually. In this way, it is possible to plan a digital model, a digital shadow or a digital twin. The background of the approach is actually a reference architecture model for a smart factory developed by our team and presented in detail in a previous research paper [42]. The architecture model shows crucial technologies for the operation of modern manufacturing systems.

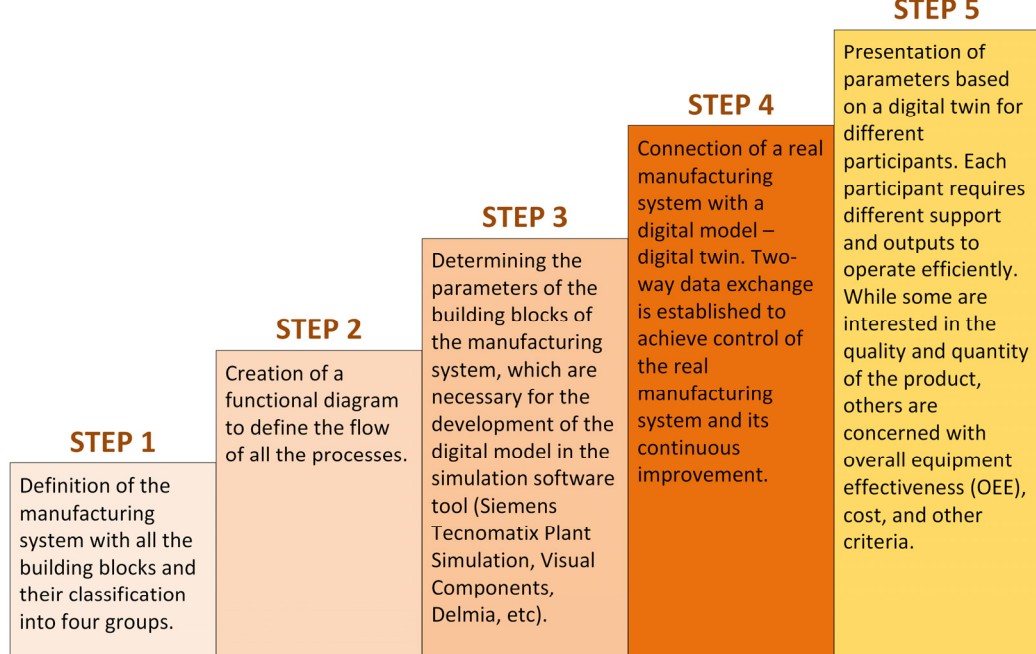

**Figure 2.** Our five-step approach that guides the users from the starting point to the usage of the digital twin.

## 2.1. Step 1

Each discrete manufacturing system (e.g., linear assembly line) consists of various building blocks (e.g., robot, assembly station, warehouse, machine vision). In the first step of the approach, the building blocks are broken down into four groups (Figure 3) (i.e., "Fabrication" building blocks, "Logistics" building blocks, "Storage" building blocks, and "Inspection" building blocks). The reason for dividing a discrete manufacturing system into four groups is that each group requires its own specific parameters and its own way of developing a digital twin. With such a classification, each building block of the manufacturing system can be placed into one of these groups. Our research and the proposed approach are carried out for the domain within the system boundary in Figure 3 and have proven useful based on previous experience in modelling and planning the digital twin as part of industrial projects [43,44].

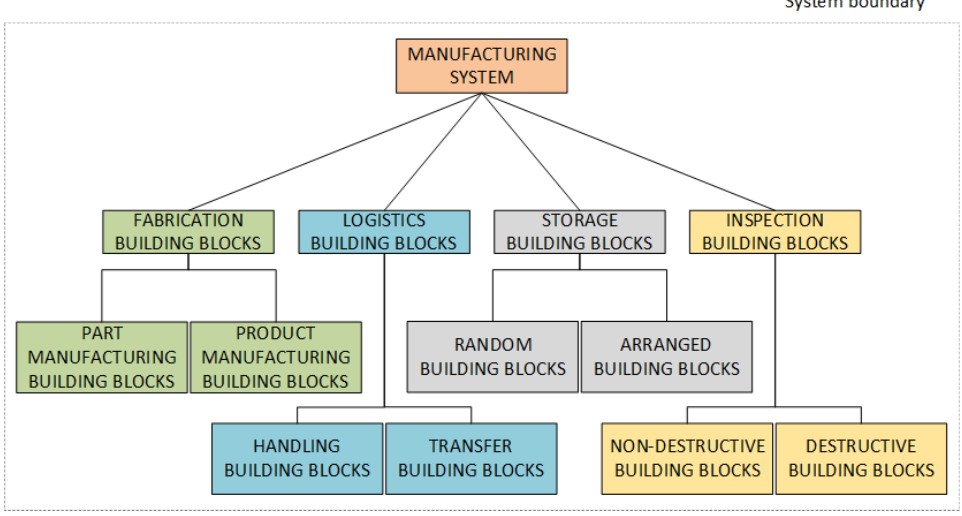

**Figure 3.** The manufacturing system broken down into four groups of building blocks.

The first group represents the building blocks for the fabrication processes. This group is characterized by the following features: the product's location remains unchanged; a new geometry of the product (or changes in the material structures: e.g., heat treatment) is generated and new information is collected. "Fabrication" building blocks consist of "Part manufacturing" and "Product manufacturing" building blocks according to the number of parts involved in the process. If one part is involved in the processes, they belong to "Part manufacturing" building blocks (the most common are building blocks for deep drawing, milling, bending etc.). If more than one part is involved in the processes, they belong to "Product manufacturing" building blocks (the most common are assembly station, disassembly station, welding building block etc.).

The second group represents the building blocks for the logistics processes and are characterized by the following features: the product's location is changed; the amount of information and the geometry/material structure of the product remains the same. Depending on the machine or who carries out the logistics process, "Logistics" building blocks can be assigned to "Handling" or "Transfer" building blocks. "Handling" building blocks have a fixed position, the product's position changes (e.g., robots, collaborative robots, various manipulators bolted to the floor, and various types of conveyors). "Transfer" building blocks can be moved around the factory floor together with the product (e.g., forklifts, drones, automated guided vehicles (AGVs)).

The third group of building blocks are "Storage" building blocks. Their characteristics are: location and geometry or material structure of the product remain the same; the amount of information remains unchanged. "Storage" building blocks can be divided into "Random" or "Arranged" building blocks depending on the orientation and position of the products. "Random" building blocks (where the products are not oriented or positioned in a specific order) are, for example: box, table, shelf storage etc. The products belonging to "Arranged" building blocks are oriented and positioned (e.g., pallet, warehouse, shelf storage).

Lastly, "Inspection" building blocks are described. This group's feature is that the product's location does not change during the process, while information is added—a correct or faulty product. The group is divided into "Non-destructive" or "Destructive" building blocks. The feature of "Non-destructive" building blocks is that during the process the geometry and the structure of the product is not changed (e.g., building blocks for visual methods, acoustic and ultrasonic methods, and radiological methods, various types of measurements (length, weight)) and even measurements and inspections performed by an operator). During the process that takes place in "Destructive" building blocks, the geometry and structure of the product is changed (e.g., building blocks for penetration methods, pull-off methods, bench scale sample testing and technical scale component testing). As a rule, the product is destroyed and is unusable for further processing.

*2.2. Step 2*

In order to develop a digital model and later a digital twin, a detailed definition of the manufacturing system with all processes and their sequences is required. This can be represented with the help of a functional diagram based on the distribution of the building blocks to the four described groups. The development of the functional diagram starts with the definition of the first process in the manufacturing system and follows with the sequence of other processes executed by the building blocks. It contains information about the building blocks that are involved in the process.

The functional diagram consists of rectangular blocks, with arrows showing the processes flow and feedback loops (Figure 4). The block represents the process that is being executed. The information can be characterized as follows:

- Inputs include information about the building blocks used to execute the process and also the information about the material and parts needed for the process.
- Outputs include output information, i.e., results of the process, and objects manufactured during the process.

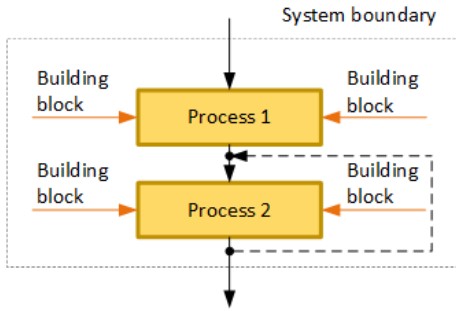

**Figure 4.** The functional diagram.

### *2.3. Step 3*

In this step, the development of a digital model is described. The development of a digital model and the definition of its parameters is the same regardless of the simulation software used for its development. Developing a digital model requires defining the specific parameters for each building block of the manufacturing system. Relevant parameters allow the correct and efficient operation of digital models, which are the basis of the extension to the digital twins by introducing the feedback control loop with the real system.

#### 2.3.1. Defining the Parameters

In order to create a digital model, it is necessary to choose only the appropriate parameters, otherwise the system becomes unmanageable [45]. For this reason, the parameters of the building blocks are defined based on the classification of the manufacturing system (as described in Step 1, Figure 3), taking into account the constraints of each building block. Since the focus of our research is the development of a digital model for logistics (i.e., material flow), it is necessary to include the parameters which directly affect the logistics process. Required parameters are process times of operations (e.g., injection, milling, assembly). The times can be recorded via embedded sensors. "Fabrication" building blocks require information about the MTTR (mean time to repair), which is defined as the average time required to repair a failed building block. Although many "Fabrication" building blocks can be found in real manufacturing systems, in this paper we define the parameters for some of them (Figure 5), which are often included in linear assembly lines (the focus of our research).

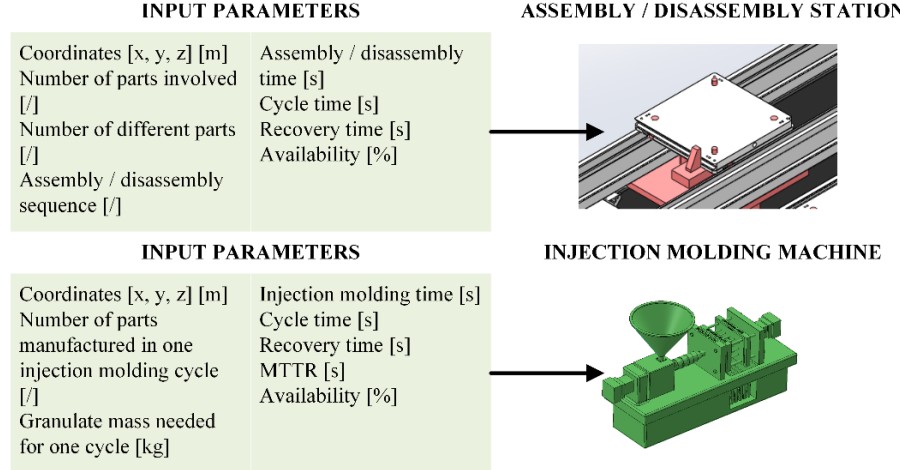

**Figure 5.** Two examples of "Fabrication" building blocks.

The important parameters for "Handling" building blocks are the coordinates, which determine the location and orientation of the building block in the manufacturing system. Other parameters include velocity, acceleration and deceleration with time, dimensions,

the number of parts involved in the process, various timing parameters as well as various constraints (e.g., velocity, acceleration, deceleration, maximum number of parts handled in a cycle, dead angle for robots). For the development of the digital model, it is necessary to take into account the availability and the current status of the building blocks, and in the case of transport of the parts by a machine, the MTTR must also be defined. "Transfer" building blocks require parameters that define the trajectory (to determine the path), path length, the velocity, acceleration and deceleration with time and the cycle time. Another set of parameters includes the number of parts transported in a cycle and various constraints that must be considered during the development of the digital model (e.g., max/min velocity, max/min acceleration and deceleration). Figure 6 shows some examples of "Logistics" building blocks.

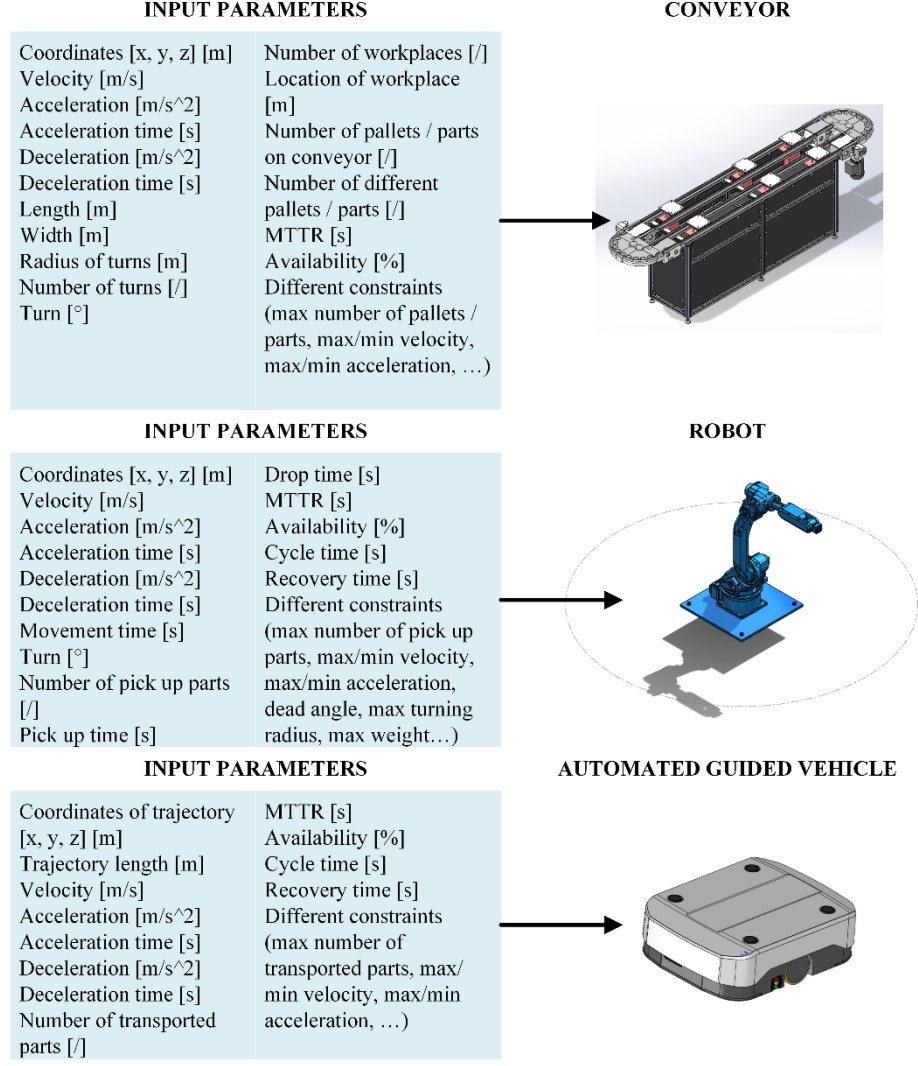

**Figure 6.** Examples of "Logistics" building blocks.

The characteristic of the "Arranged" building block is the order in which the parts are loaded and unloaded. The digital model requires the information about the unoccupied locations and where the next part should be stored. "Arranged" building blocks are limited by both the number of locations and the size of the parts. "Random" building blocks are characterized by the fact that loading and unloading is completely unordered and the parts have no predefined location. The constraints are the volume of the "Random" building block or the maximum allowable weight of the full building block. This information is needed, for example, to determine the time of unloading the stored parts. Figure 7 shows the parameters for an "Arranged" and a "Random" building block.

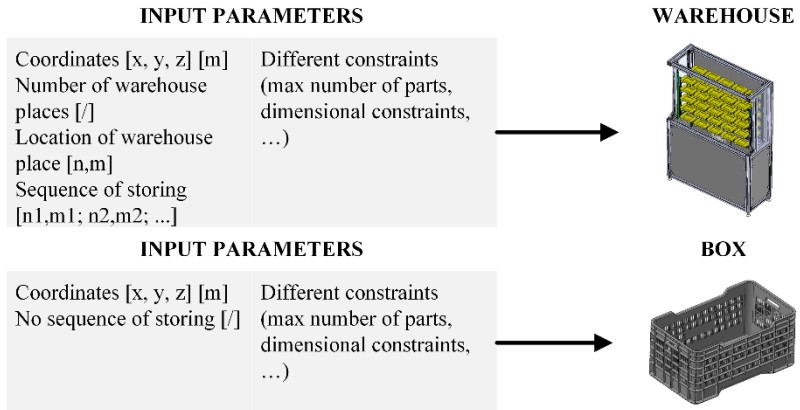

**Figure 7.** Examples of "Storage" building blocks.

The group of "Inspection" building blocks requires information about the duration of the inspection process (i.e., cycle time) and the estimated percentage of correct parts. If a device is used for inspection (e.g., camera, strength test device), it is also necessary to define the availability of the device and the MTTR. These two parameters are not required if the inspection is performed by an operator. Figure 8 shows a list of parameters for machine vision.

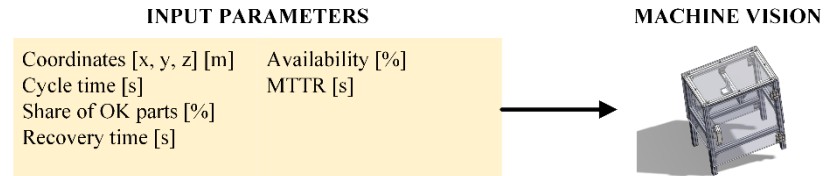

**Figure 8.** Example of "Inspection" building blocks.

The digital model of all interconnected building blocks requires general information for its operation. The information relates to the quantity of raw materials or parts, customer orders, manufacturing schedule and resources. The schedule includes the number of shifts, time of shift operation, breaks, time of shift start and shift end.

### 2.3.2. Developing the Digital Model

The procedure for digital model development has already been presented several times in the literature [46–48]. The most important points will be reviewed here. The procedure is very similar regardless of the software tools used (Siemens Tecnomatix Plant Simulation, Visual Components, Delmia, etc.).

The development of a digital model begins with the definition of the manufacturing system layout for which we want to develop a digital model. The layout includes the dimensions, the types of building blocks, and the location and orientation of the building blocks in the system. For all the building blocks of the manufacturing system, we need to determine the parameters defined in Section 2.3.1. Depending on the simulation software tool used, certain parameter values can be entered in graphical interfaces for building blocks in the program library; for other non-standard building blocks (which are not included in the library and are usually more complex), it is necessary to generate program code.

After the parameters have been inserted, the building blocks are connected in the order defined in Step 2. As with parameters, certain simple dependencies are executed with graphical interfaces, and for more complex dependencies it is necessary to generate program code with mathematical functions. After the development of the digital model is complete, the verification and validation of the digital model follows. This must be done to ensure that the digital model works exactly like the real manufacturing system.

### 2.4. Step 4

In this step, the connection between the digital model and the real system is presented. The digital model collects and analyzes data from sensors installed in the real manufacturing system and performs the simulation run, for example, to optimize the sequence of manufacturing orders. The results of the simulation are then transferred back to the real system via the feedback control loop in the form of control sequences. The real system adapts while the digital model again collects and analyzes sensor data and, if necessary, corrects the input parameters of the simulation [49]. With such connectivity, a digital twin is created.

Figure 9 shows the idea of the connection between the physical and digital worlds. The physical world shows the evolution of the manufacturing system through time, as its state changes through the process from "State of manufacturing system at time t" to "State of manufacturing system at time t + 1". Monitoring these states enables the development of "Digital model of the manufacturing system at time t". Due to the constant changes in the real system, monitoring must be possible in realtime. That way the model always reflects the actual state of the real system. To make this possible, simulations are performed continuously. The result of the simulation and analysis is "Digital model of the manufacturing system at time t + 1". This ensures the adaptation of the process of the existing real manufacturing system "State of manufacturing system at time t + 1" to the new form "State of manufacturing system at time t + 2". Appropriate interfaces are included to connect the digital and physical worlds.

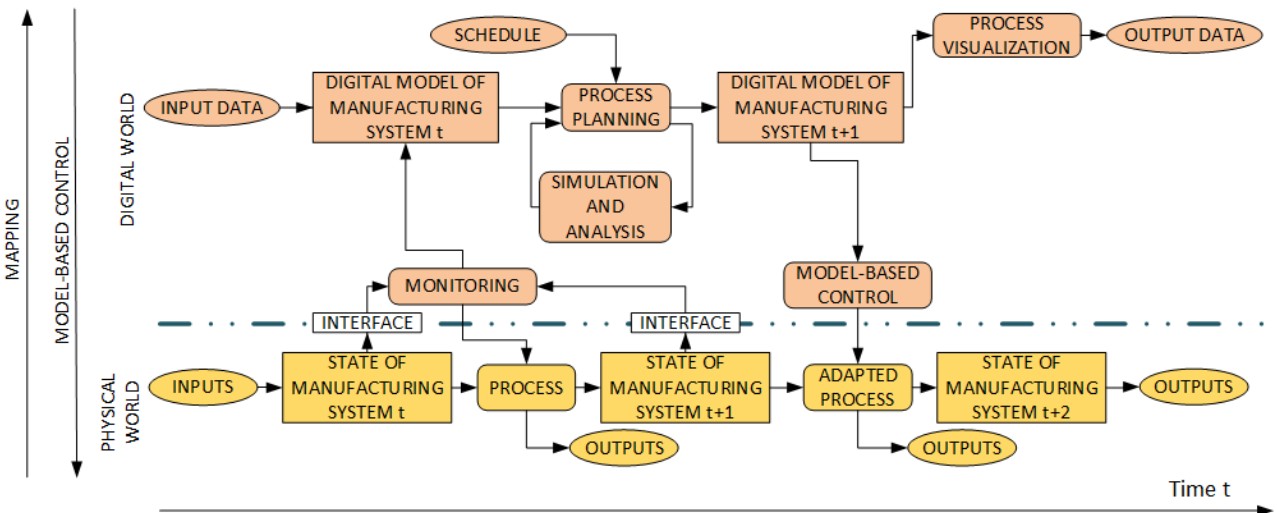

**Figure 9.** Position of the model and the connections between the physical and digital worlds.

Simple logical dependencies of processes are enumerated with the software tool's built-in functions, and digital agents are used for more complex logical dependencies (mathematical algorithms). Each digital agent performs a specific task and operates with local data. Digital agents upgrade and enrich the digital twin and increase performance by using local data. Digital agents receive data from the digital twin and suggest the best solution based on the received data. Here is the list of digital agents that have been developed to upgrade the digital model to the digital twin (all agents are thoroughly described in the PhD thesis [50]):

- Digital Agent of Initial state (D.A.I) is a digital agent that sets the correct initial state of the manufacturing system in the digital twin (e.g., warehouse has 20 locations for base parts and 25 locations for finished products; there are 3 base parts and 7 finished products in the warehouse at specific locations).
- Digital Agent of Orders (D.A.O.) is a digital agent that determines which orders can be manufactured based on the current state of the system checks.

- Digital Agent of Manufacturing (D.A.M.) is a digital agent, which determines the sub-optimal manufacturing plan i.e., the sequence of assembly operations for each order and which building blocks will perform assembly operations.
- Digital Agents of Jobs (D.A.J.) is a digital agent that generates all jobs for the robots and coordinates the sequence of jobs.
- Digital Agent of Disturbance (D.A.D.) is a digital agent responsible for verifying and resolving unplanned disturbances; the agent detects the disturbance, checks for possible solutions, and, if possible, offers a solution to resolve it.
- Digital Agent of the Intelligent Algorithm (D.A.A.) is a digital agent with an expert system designed to optimize the schedule using an intelligent algorithm.
- Global Digital Agent (G.D.A.) is a global digital agent that coordinates and monitors the operation of all digital agents and ensures the correct flow of communication.

A substantial advantage of a connected real and virtual world is the constant exchange of information and the adaptation of a real system based on the digital twin. To achieve such connectivity, various sensors need to be installed in the real manufacturing system. The type of sensors depends on the parameters that need to be exchanged. The locations of the sensors vary; in a real manufacturing system, they are usually installed in front of and behind the building block. In such installation of sensors, the relevant product data is obtained from the real system before entering the process and after the process has been completed.

The advantage of combining the physical and digital worlds, in addition to controlling a real manufacturing system, is the digital traceability of the product (an example from the literature is presented in [51]). Digital traceability allows tracking the location of products when it is not possible to apply a bar code or RFID tag due to restrictions (e.g., geometry, size).

### 2.5. Step 5

The digital twin enables the representation of different information for different participants (i.e., operators, developers, maintenance personnel, planners, customers, managers) in the manufacturing system. The digital twin is the same for all participants, the difference being in the representation of the parameters. Each participant needs different support and outputs to work efficiently. In the literature, we can find various studies on parameters that need to be visualized for different participants [52,53]. Some are interested in the quality and quantity of products, bottlenecks, maintenance times, overall equipment effectiveness (OEE), while others are interested in the dimensions of the manufacturing line, the number of robots, etc. For example, management needs information about the cost of the process, while developers need more detailed information about the manufacturing processes and systems. Various computer programs can be used to visualize the parameters, such as myPRO, The Control Web System, Power BI, SIMATIC WinCC, CODESYS Visualisation, GP-PRO EX HMI, Ignition SCADA, Ansys SCADE display, etc.

The digital twin helps operators predict when raw material will be loaded into the buffer as well as other activities. Individual machine working time is important to operators because it allows them to remove full pallets without inspection. Moreover, they can inspect the products e.g., at the end of the manufacturing process.

Planners need information and statistics about all machines and processes in a manufacturing line. For a smooth process, planners use the digital twin of the manufacturing lines as a support. With the help of the digital twin, they can get machine information, determine the occupancy of the devices and workers, possible breakdowns, and the number of parts manufactured. They can also predict the working time, setup time, waiting time, and blocked time. Another benefit is creating a work schedule for the next day, the next week or for a longer period in advance in a virtual environment. Depending on the occupancy of the operators, the planners can schedule their work.

## 3. Case Study

The presented approach was validated in real environment during the development of Demo Center Smart Factory at the University of Ljubljana, which includes various manufacturing processes and systems (Figure 10). It was examined whether the developed approach can simplify the process of developing digital models and digital twins, with the aim of not having to redefine the parameters in later development phases. The approach was used to verify the proper functioning of the system when the user follows the proposed five steps.

The manufacturing system consists of various building blocks (I) a conveyor, (II) four pallets (with RFID tag), (III) a warehouse for base parts and finished products (WBPFP) with 45 locations (4 × 5 locations in the first row and 5 × 5 locations in the second row) where one part of the warehouse stores base parts (first row) and the other part stores finished products (second row), two industrial robots ((IV) Robot 1 and (V) Robot 2) with different payload capacities, (VI) a warehouse for robot grippers (WRG), (VII) three buffers along the conveyor, each containing 64 tiles of the same color (red, green or blue), (VIII) box for faulty finished products, (IX) machine vision, and (X and XI) two automated assembly stations (equipped with RFID reader). Each building block has its own specifications and tasks. The final product is a mosaic assembled into various patterns and colors. Such a product is not used for an industrial case, but it is very suitable for research, as it allows various combinations and thus requires a flexible manufacturing system. In the case study, 12 products (Table 1) in 4 successive scenarios are assembled.

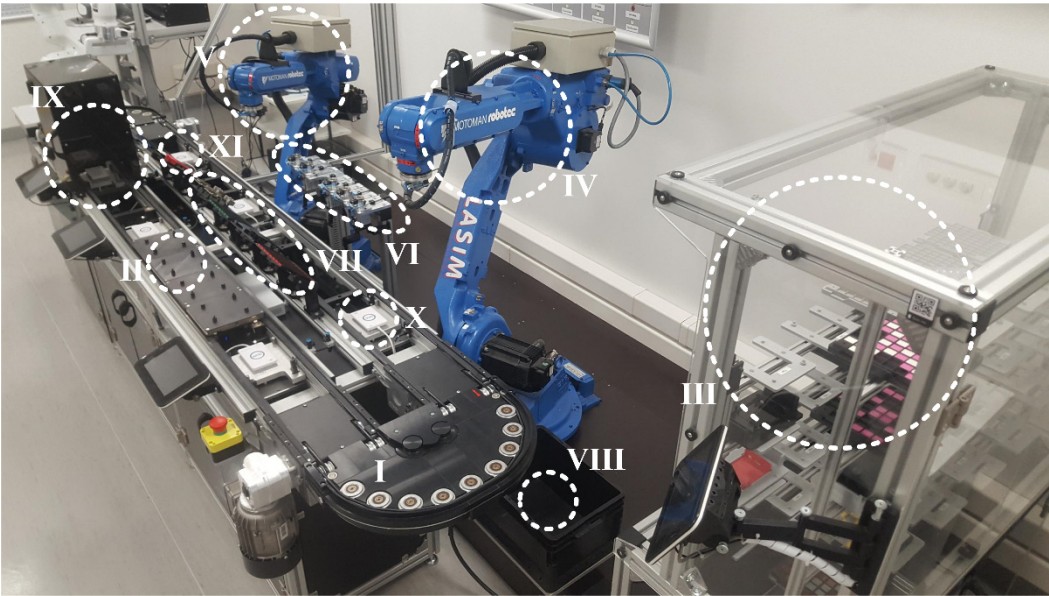

**Figure 10.** The Demo Center Smart Factory.

The process of assembling a product is as follows. It is assumed that the empty pallet (Figure 11a) is on the conveyor at the Assembly station 1 (Figure 10, X). Robot 1 picks up the appropriate gripper for the base part from the warehouse for the robot grippers (WRG). Then it moves to the first row of the warehouse for base parts and finished products (WBPFP), takes the base part from it (the base part has 64 locations for tiles and is symmetrical), transfers it to the Assembly station 1 and positions it onto the empty pallet (Figure 11b). After positioning is complete, product assembly begins. If the product contains red or green tiles, assembly begins at this assembly station (Assembly station 1). Robot 1 moves to the WRG and picks up the appropriate gripper for the tiles. Then Robot 1 begins assembling the product in the specified order. The order of assembly is determined using the digital twin of the manufacturing system and the associated algorithms and digital agents. The tiles are stored and ordered in buffers next to the conveyor. After the

assembly of the red or green tiles, the pallet moves to the Assembly station 2 (if blue tiles are required to complete the order) or to the machine vision if the assembly was finished on the Assembly station 1.

**Table 1.** 12 products assembled in 4 successive scenarios with different tile color and quantity.

| Product | | Number of Tiles | | |
| :---: | :---: | :---: | :---: | :---: |
| | | **Red Color** | **Green Color** | **Blue Color** |
| 1 (Figure 11c) | | 4 | 4 | 4 |
| 2 (Figure 11d) | Scenario 1 | | 16 | |
| 3 (Figure 11e) | | | | 14 |
| 4 (Figure 11f) | Scenario 2 | 6 | 4 | 4 |
| 5 (Figure 11g) | | | | 11 |
| 6 (Figure 11h) | | 16 | | |
| 7 (Figure 11i) | Scenario 3 | | 14 | |
| 8 (Figure 11j) | | | | 10 |
| 9 (Figure 11k) | | | | 14 |
| 10 (Figure 11l) | | | 11 | |
| 11 (Figure 11m) | Scenario 4 | 16 | | |
| 12 | | | | 19 |

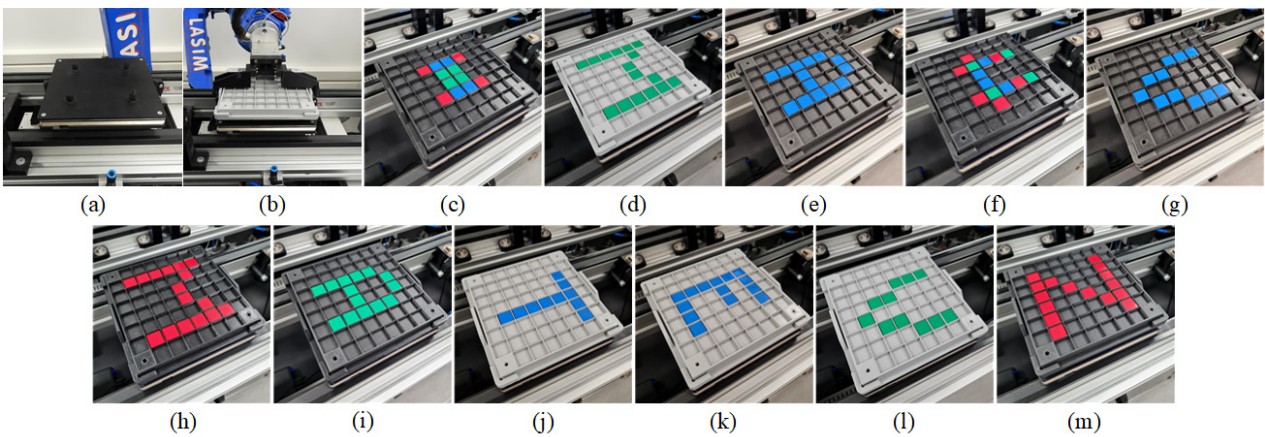

**Figure 11.** (**a**) Empty pallet; (**b**) Pallet with a base part; (**c**) Product 1; (**d**) Product 2; (**e**) Product 3; (**f**) Product 4; (**g**) Product 5; (**h**) Product 6; (**i**) Product 7; (**j**) Product 8; (**k**) Product 9; (**l**) Product 10; (**m**) Product 11.

Assembly continues with Robot 2, which assembles the final product. The finished product is transported via the conveyor to the machine vision. Here it is checked whether the finished product is correctly assembled (colors, pattern). The information about the correctness of the assembly is important for the next processes in the manufacturing system, such as the control of Robot 1 when manipulating the finished product, the work of the digital twin, an order sequence, etc. The product is transported along the conveyor and comes back to the Assembly station 1. Robot 1 picks up the appropriate gripper and transfers the correct finished product (OK) to the second row of the WBPFP based on the information from machine vision and the digital twin. Faulty finished products (NOK) are removed from the process and put into a box for faulty products.

## 4. Results and Discussion

In this section, the results of the digital twin developing approach (described in Section 2) are presented using a case study as an example. The results are presented in the form of tables, diagrams and figures and commented for each of the five steps.

## 4.1. Step 1

The case study consists of two "Fabrication" building blocks, three "Logistics" building blocks, five "Storage" building blocks, and one "Inspection" building block (Table 2).

**Table 2.** Building blocks of the manufacturing system divided into the four groups.

| "Fabrication" Building Blocks | "Logistics" Building Blocks | "Storage" Building Blocks | "Inspection" Building Blocks |
|---|---|---|---|
| Assembly station 1 (X) | Conveyor (I) | Pallet (II) | Machine vision (IX) |
| Assembly station 2 (XI) | Robot 1 (IV) | Warehouse for base parts and finished products (III) | |
| | Robot 2 (V) | Warehouse for robot grippers (VI) | |
| | | Buffer for tiles (VII) | |
| | | Box (VIII) | |

## 4.2. Step 2

Figure 12 shows the sequence of processes in the case study necessary for the development of the digital model in the form of a functional diagram.

## 4.3. Step 3

The third step is the development of the digital model of the presented case study. In our case, the Siemens Tecnomatix Plant Simulation 14.2 software package was used to develop the digital model. The development started with the definition of the layout of the manufacturing system with all the building blocks. Each type of a building block requires specific parameters to define it. The defined parameters with their values for the conveyor building block are described in the text below and in Figure 13.

The name of the conveyor is required both to identify the building block (e.g., Conveyor1, Assembly_station1) and for more transparent planning. To determine the correct position and orientation of the conveyor, the tangential angle $\alpha$ in degrees is required ($\alpha = -180°$ from the starting point).

The next parameters are the length (L1 = 2.57 m; L2 = 0.07 m) and the width (W = 0.16 m) of the conveyor, the number of curves (4), the angle of curves (90°) and the radius (R = 0.2 m) of curves. The required data also includes velocity (0.15 m/s), acceleration (0 m/s$^2$) and deceleration (0 m/s$^2$) of the conveyor.

The information on the number of workstations and their locations is also required. The workstation is defined as the location where the product is involved in a process. In our case study, there are 3 workstations. The location of the workstation is defined as the distance from the starting point (a = 0.45 m; b = 1.67 m; c = 0.45 m). It is also important to have information about the number of different pallets or parts that are transported by the conveyor. In the case study, there are 4 pallets with the same dimensions.

The availability of the machine is proportional to the time that the machine is in an operational state, represented by a value between 0 and 100 percent; in our case study, the availability is 100%. The next parameter is MTTR—a measure of maintainability. It is defined as the average time to repair a machine and is 0 s in our case study.

The procedure for defining the parameters needed for other building blocks is the same as for the conveyor building block. Then the development of the digital model begins. For the development of the digital model, we followed the procedure described in Section 2.3.2.

Mathematical methods and logical dependencies are used to connect the digital models of the building blocks to the digital model of the entire manufacturing system in

the Siemens Plant Simulation software tool, using the SimTalk programming language. The processes in the digital model should follow each other as defined in Step 2.

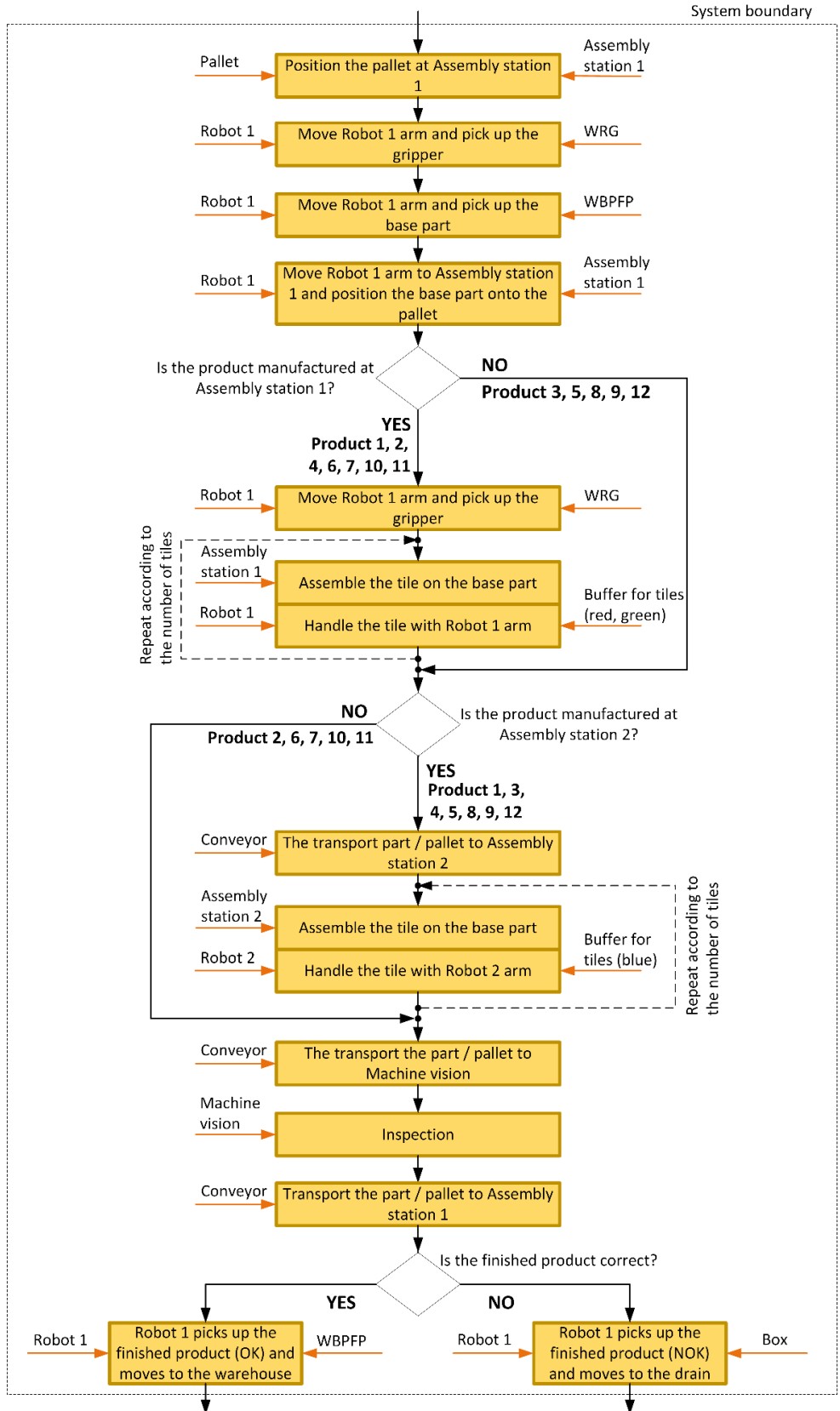

**Figure 12.** The sequence of processes in the manufacturing system case study.

| Name: Conveyor1<br>Orientation from starting point: -180 °<br>Length [m]:<br>L1 = 2.57<br>L2 = 0.07<br>Width [m]:<br>W=0.16<br>Number of turns: 4<br>Radius of the turn [m]:<br>R − 0.20<br>Angle of turn [°]:<br>90 | Velocity [m/s]: 0.15 m/s<br>Acceleration [m/s^2]: 0<br>Time of acceleration [s]:<br>0<br>Deceleration [m/s^2]: 0<br>Time of deceleration [s]:<br>0 | Number of workplaces: 3<br>Location of workplace [m]:<br>a = 0.45<br>b = 1.67<br>c = 0.45<br>Number of pallets: 4<br>Number of different pallet: 1<br>Availability [%]: 100<br>MTTR [s]: 0 | Constraints:<br>Max velocity [m/s]: 0.2<br>Max acceleration [m/s^2]:<br>0.5 |
|---|---|---|---|

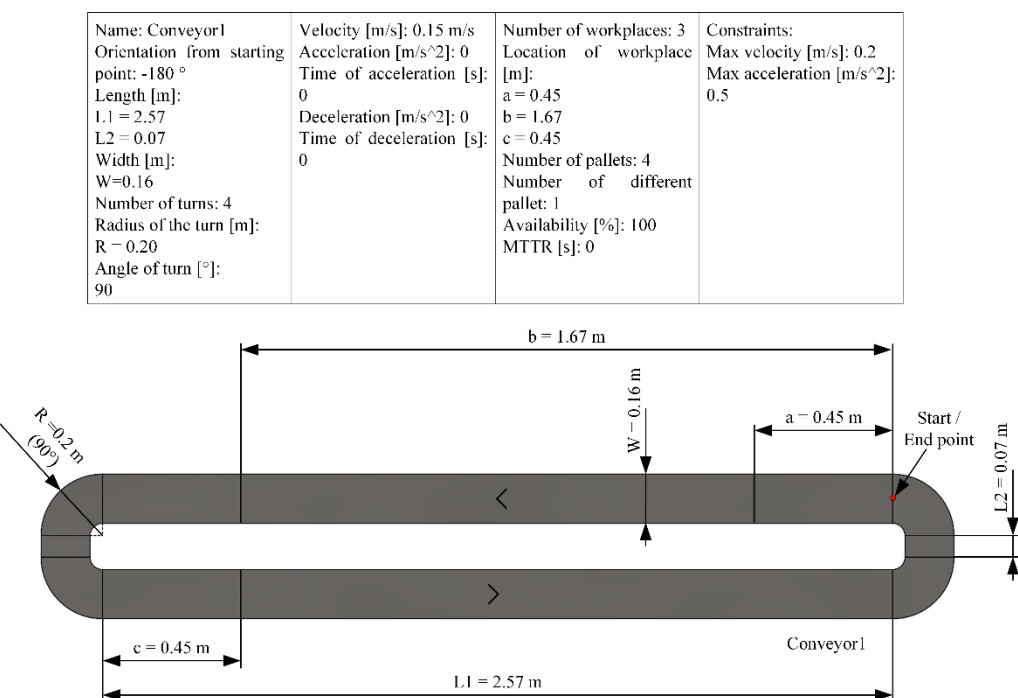

**Figure 13.** Values used for the digital model of the conveyor.

The result is a functional digital model with all the details of a real system that functions exactly like a real manufacturing system (Figure 14). The feedback loop is not yet established in this step, which means that if there is a change in the real world, the digital model will not change accordingly. The vertical red lines in Figure 14 show the virtual sensors that detect the beginning and the end of the process, which are crucial for matching a digital model to a real system. Other sensors installed to match the digital model to the real system include a machine vision system to detect the quantity and colors of tiles in the buffers, an RFID reader in the WBPFP, and proximity (inductive) sensors in the WRG.

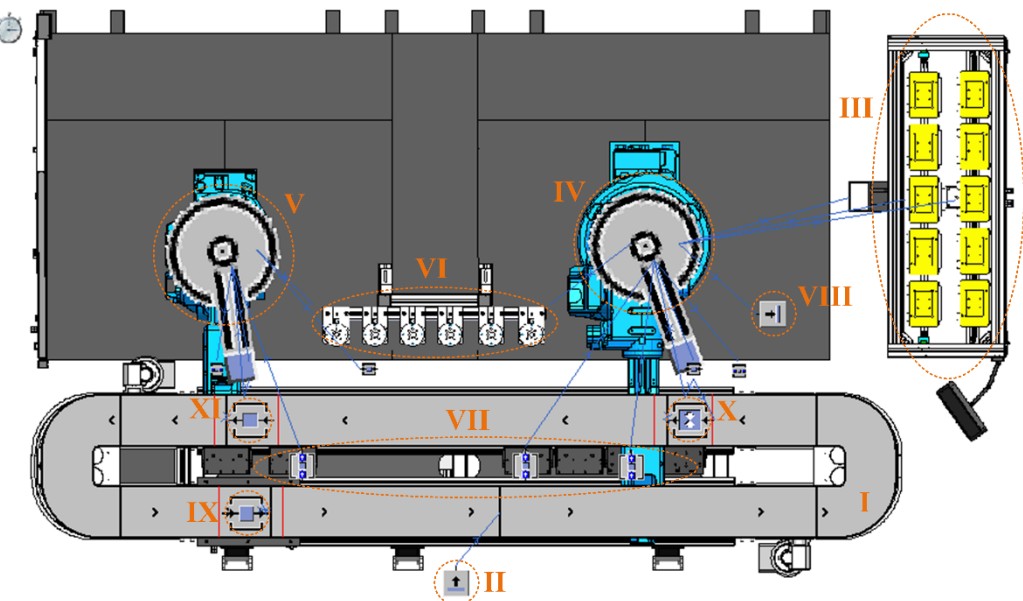

**Figure 14.** The digital model of the manufacturing system with all the building blocks and connections between them.

### 4.4. Step 4

In the fourth step, the actual digital twin of the process is created. The digital model presented in Step 3 is connected to the experimental setup of the Demo Center Smart Factory (as indicated in the beginning of Section 3) via a feedback control loop (shown in Figure 15 with bi-directional data flows between the digital model and the six main building blocks). The six main building blocks of the experimental setup represented in Figure 15 are: (V) Robot 1 and (IV) Robot 2, (X) Assembly station 1, (XI) Assembly station 2, (IX) machine vision, and (III) WBPFP. They are all equipped with separate controllers.

The feedback control loop is implemented in two steps: (i) the digital twin in connection with digital agents (described in Section 2.4) collects data about the current state of the manufacturing system (e.g., quantity and type of available base parts, assembly parts, availability of robot grippers . . . ) and updates the input data for the simulation (e.g., optimal order sequence). After the simulation run, the order sequence and operations to be performed are transmitted from the digital twin to the appropriate building blocks. For example, information about the current order and the assembly operations to be performed are transmitted to the assembly stations. Information about robot jobs to be executed as well as generated tasks are transmitted to robots. (ii) Each building block filters locally acquired data during the assembly process (e.g., data about orders detected by RFID, assembly station status, and pallet presence from proximity sensors, product quality check from machine vision) and transmits it back to the digital twin for continuous (re-)validation of simulation results.

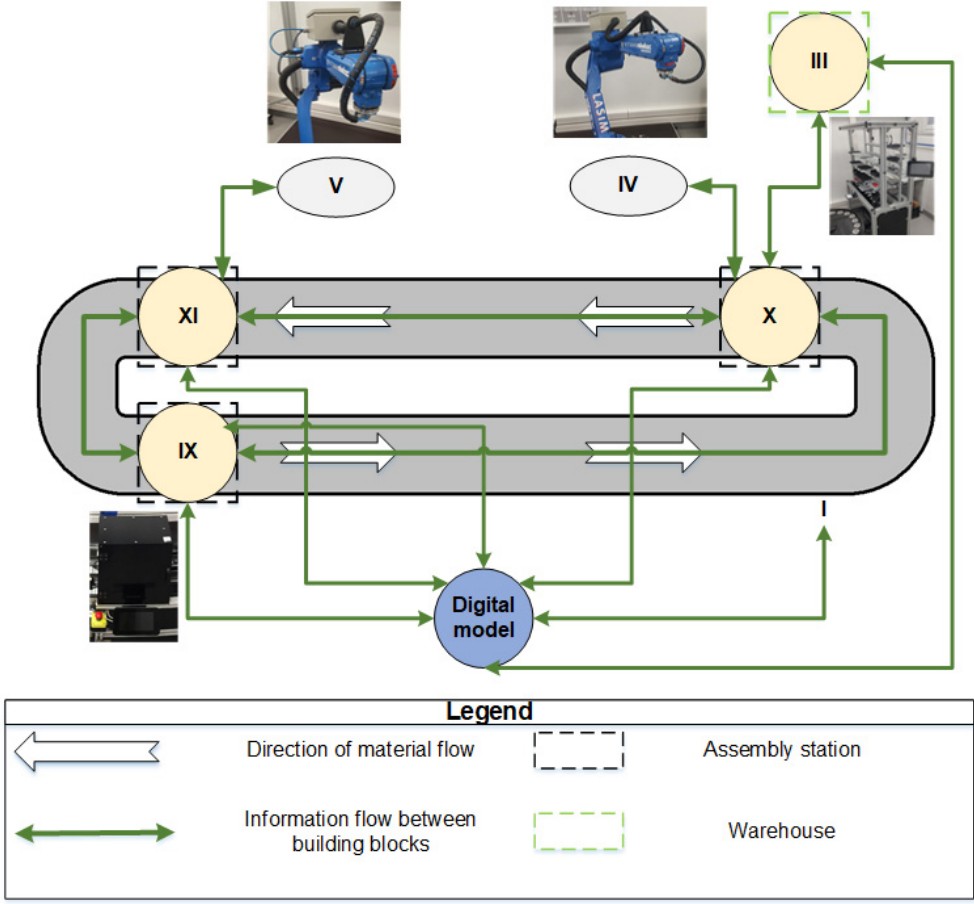

**Figure 15.** Connections between the digital model and the six main building blocks in the case study.

The described steps (i) and (ii) for implementing the feedback loop are now presented in the case study. A camera mounted on the ceiling of the demo center checks the current states of the tiles in buffers. The availability of the robot grippers is checked by proximity

sensors in the WRG. The availability of the base parts in the WBPFP is acquired by checking RFID data from the WBPFP controller. Gathered states of the parts are sent to the digital twin before the scenario is executed. The digital twin receives information that there are 11 base parts and 0 finished products in the WBPFP. It also receives the information that there are 3 base parts in the second column and 4 base parts in each of the other two columns. There are 64 blue, green and red parts (tiles) in the buffers (Figure 16a).

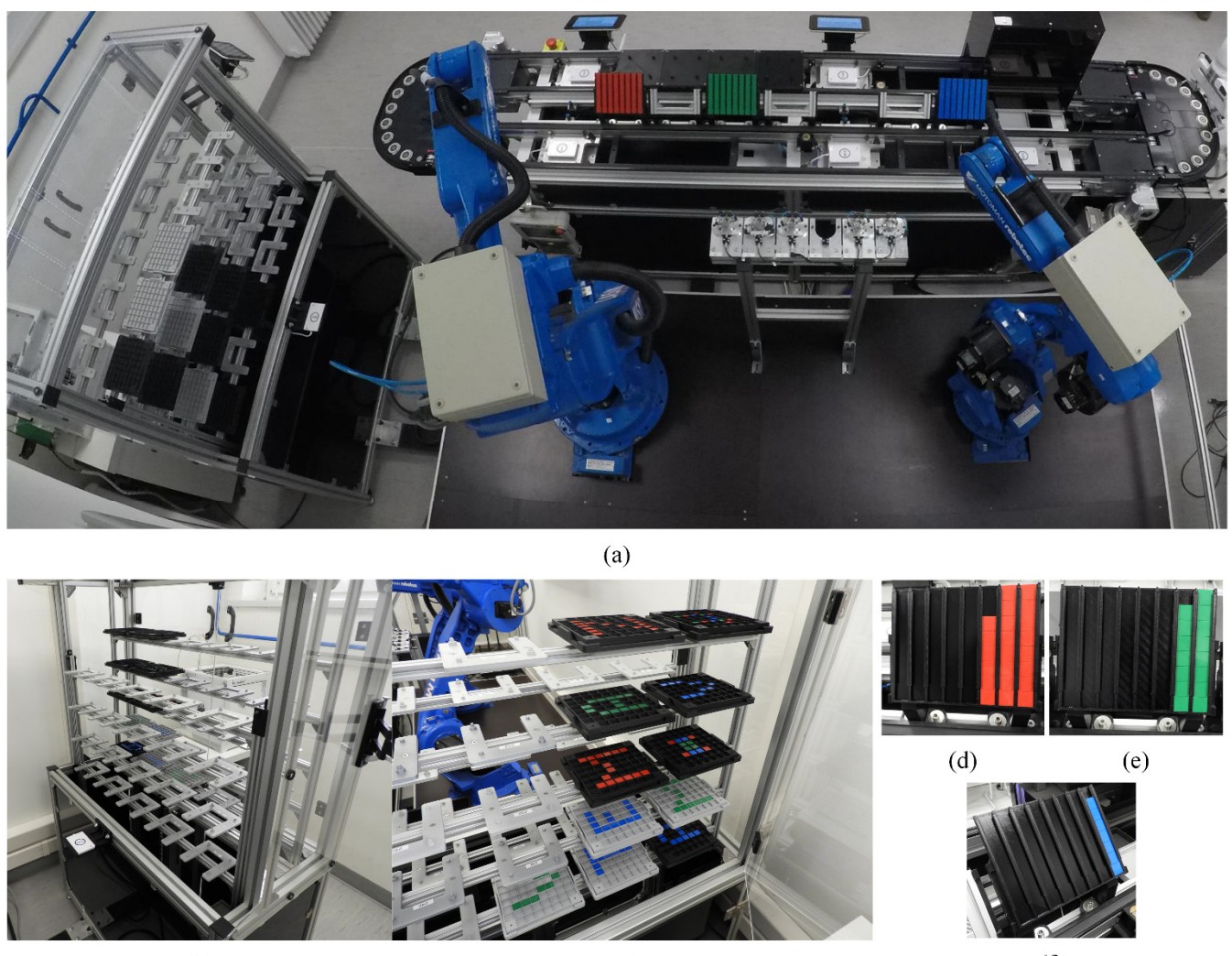

**Figure 16.** The figure shows: (**a**) the state of the assembly system before the start of assembly with full WBPFP and buffers; (**b**) no base parts in the first row of the WBPFP after assembling; (**c**) finished products in the warehouse in the second row; (**d**) the state of the buffer with red tiles, (**e**) green tiles, and (**f**) blue tiles after the completion if the assembly process.

Based on the state of the WBPFP and WRG and the buffers, as well as the products that need to be assembled in a given scenario, the Flip and Insert (FI) algorithm (the algorithm is described in detail in the PhD thesis of a member of our team [50]) computes in the digital twin the suboptimal order sequence for the assembly of the products (assembly schedule) in Scenario 1. The assembly schedule is sent to the controllers in the real system and the assembly of Scenario 1 begins. Immediately after the completion of Scenario 1, the sensors check the availability of the parts in the WBPFP, WRG and buffers and provide information to the digital twin. For Scenario 2, the FI algorithm again computes the suboptimal order sequence. The same process is repeated for all scenarios. The last (fourth) scenario involves the assembly of three products. The number of parts before the last scenario is as follows: 2 base parts and 9 finished products in the WBPFP and 7 blue, 26 green and 38 red tiles in

buffers. For the assembly of Product 12, 19 blue tiles are needed. With the help of the digital agent, the digital twin receives information from the real system that it is not possible to assemble Product 12, but it can assemble the other two products from Scenario 4.

For the remaining two products, the FI algorithm computes the best order sequence and sends the information to the real system. Figure 16b,c show the state in the WBPFP and buffers after all scenarios are completed. The WBPFP with base parts is empty, and there are 11 finished products in the second row. There are still 21 red tiles (Figure 16d), 16 green (Figure 16e) and 7 blue tiles (parts) (Figure 16f) in the buffers.

Table 3 shows the order sequence proposed by the digital twin using the FI algorithm. Other results refer to the machine vision building block, where each product is inspected. The times required to inspect the product in a simulation environment (digital twin) and the times required to inspect the product in a real system are also shown. From the small difference between the simulation and real times, it can be concluded that the digital twin is functioning properly and is properly validated. Machine vision sends information about the suitability of the product to the digital twin, and the digital twin informs the real system whether the product should go into the WBPFP or into the box. In the case study, all products were correct, only product 12 could not be manufactured due to missing blue tiles.

**Table 3.** Collected data from the digital twin showing the order sequence, the duration of the inspection process in the digital and real environment, and information about the correctness of the finished product.

| Order Sequence | Product | Simulation Time [s] | Machine Vision Real Time [s] | Information |
|---|---|---|---|---|
| 1 | 3 | 16 | 16 | Product is OK |
| 2 | 2 | 17 | 16 | Product is OK |
| 3 | 1 | 16 | 16 | Product is OK |
| 4 | 5 | 17 | 16 | Product is OK |
| 5 | 4 | 17 | 16 | Product is OK |
| 6 | 8 | 16 | 16 | Product is OK |
| 7 | 9 | 16 | 16 | Product is OK |
| 8 | 6 | 16 | 16 | Product is OK |
| 9 | 7 | 16 | 16 | Product is OK |
| 10 | 11 | 17 | 15 | Product is OK |
| 11 | 10 | 17 | 17 | Product is OK |
| 12 | 12 | / | / | NO product |

The Figure 17 shows how the digital twin works for the case study. In the fourth scenario, the digital twin lists which products it can assemble ("CAN MAKE" in the green box in Figure 17) and which products it cannot assemble ("CANNOT MAKE" in the red box in Figure 17). The digital twin displays a table with "TRUE" values for products that can be assembled and "FALSE" values for products that cannot be assembled. In the case of Product 12, due to the missing blue tiles, the value "FALSE" and the reason why it cannot be assembled are displayed in the table (lower left side of Figure 17).

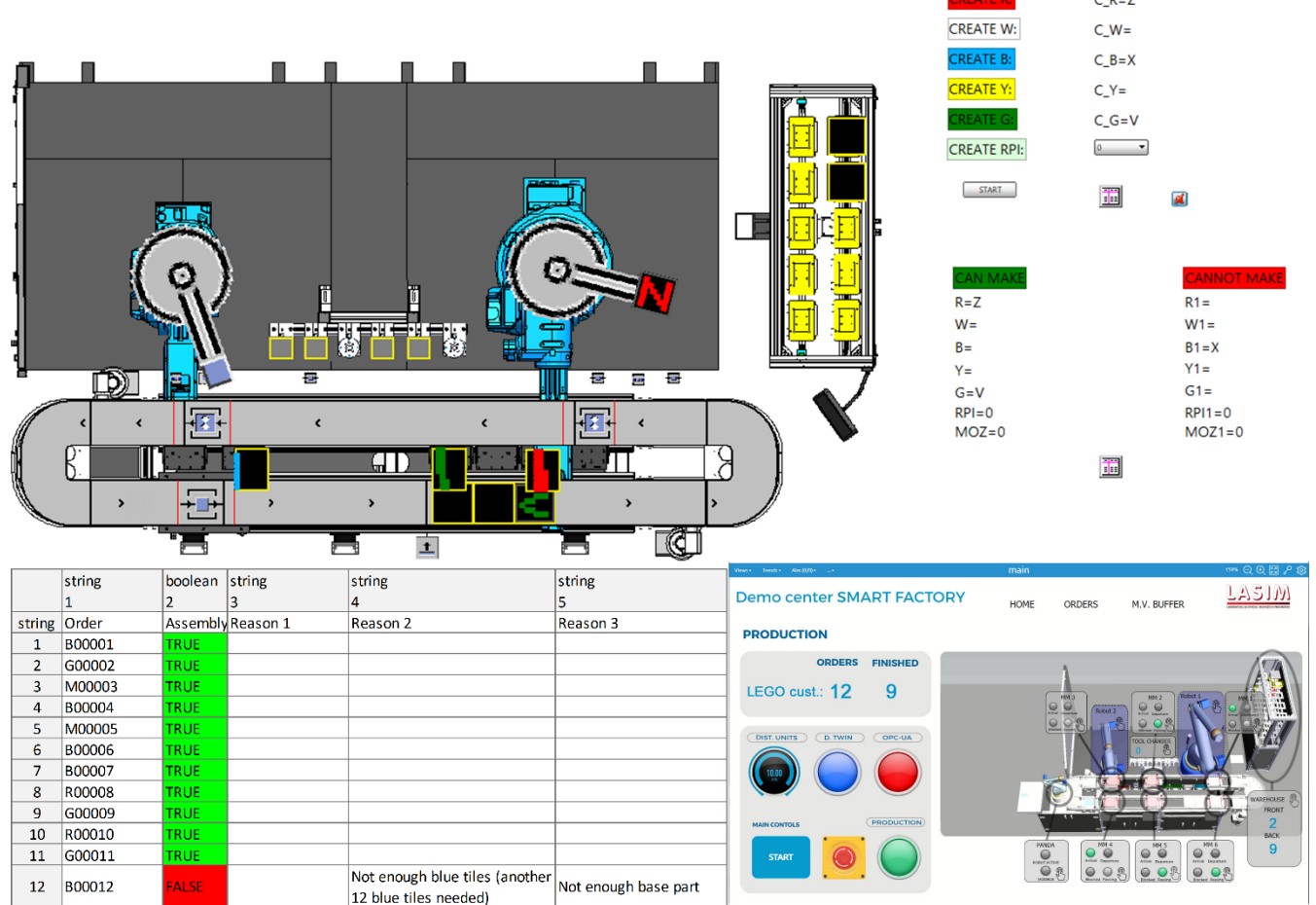

**Figure 17.** The digital twin of the Demo Center Smart Factory.

*4.5. Step 5*

During each process, the values of the process parameters are collected and stored. Each of the three workstations and WBPFP is equipped with a separate controller (Raspberry Pi 3B+, manufactured by Sony UK Technology Centre, Pencoed, UK, with Raspbian GNU/Linux 9.6, manufactured by Raspberry Pi Foundation, Cambridge, UK) which enables a distributed control structure. Controllers are connected to the sensors in its vicinity (i.e., inductive sensors near the specific workstation, RFID reader at the specific workstation/WBPFP, machine vision system connected to the controller of workstation 3). Therefore, local data storage and analysis as well as local decision-making and control is enabled. Data is stored in a local SQLite database, data analysis and local decision-making is performed by a decision algorithm that was programmed in Python (v3.7). Controllers communicate bi-directionally through an Ethernet based network. We used the OPC UA communication protocol to transfer data between the controllers. Specifically, this means that each of the controllers is also equipped with a server and a client, which were also programmed in Python.

The SCADA software we used in our work (myPRO software package from mySCADA Technologies), installed on the PC with simulation software (Siemens Tecnomatix Plant Simulation) that acts as a digital twin, allows establishing an OPC UA connection with the servers of each of the controllers. This means that the data in the SCADA system is aggregated directly from the process. The visualization display can be accessed from various devices (mobile phones, tablets, computer screens).

Results are displayed for two participants, an operator and a planner. The operator has four touch screens on which the state of the assembly system can be checked (Figure 18).

The visualization shows data from sensors in the real manufacturing system combined with data coming from a digital twin. Figure 18a shows the current machine vision process. The green lights illustrate the correct operation of the device, and red lights appear in case of a malfunction. The display shows the start time, the end time and the information about the correctness of the product; there are two states: "Product is OK" or "Product is NOK". The green lights in Figure 18b,c show the same information as described in Figure 18a. Figure 18d shows the state of the WBPFP. The green rectangles show the position of the finished products in the second row of the WBPFP. Using the digital twin, the operator can check the number of tiles, base parts, and finished products in the buffers or in the warehouse and determine the optimal loading/unloading time.

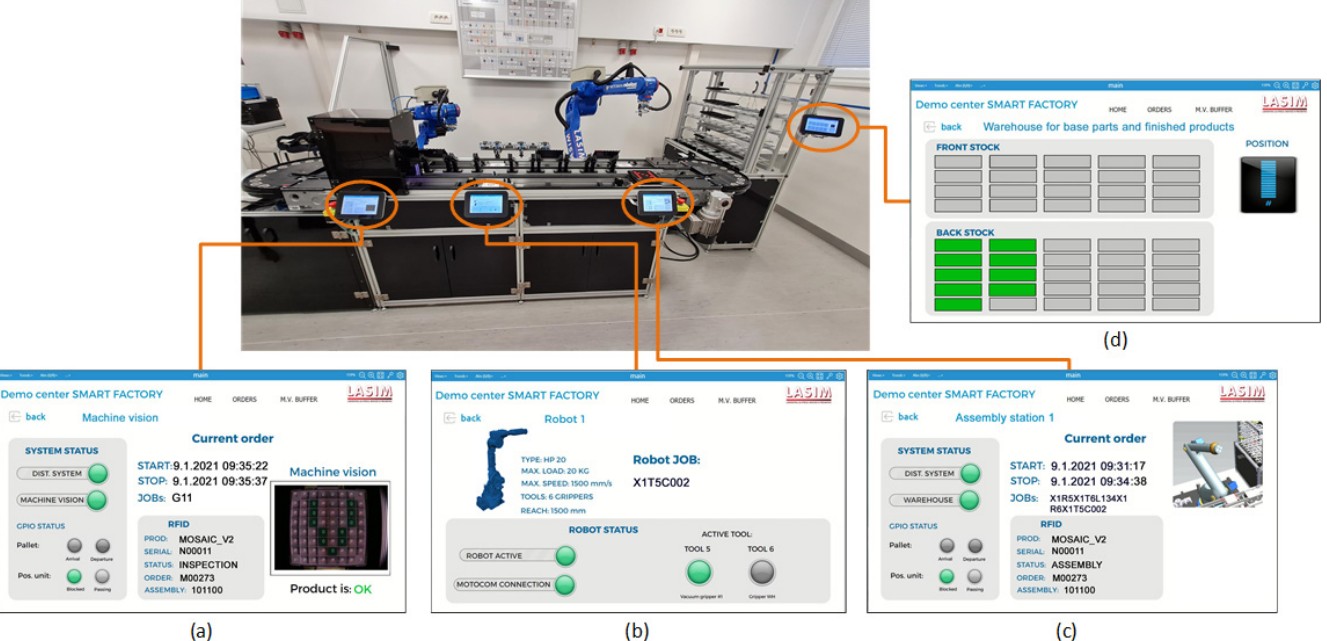

**Figure 18.** Visualization of the processes for the case study for different building blocks in the fourth scenario: (**a**) Machine vision, (**b**) Robot 1, (**c**) Assembly station 1, and (**d**) WBPFP.

The planner can use the digital twin to improve the working process by creating a suboptimal manufacturing plan using the FI algorithm. The case study shows what the optimal manufacturing plan looks like for each scenario (as a function of time and the number of parts in the WBPFP and buffers). It allows the planner to quickly adjust the manufacturing plan before starting to manufacture the products in a real system. Another benefit of using the digital twin for the planner is the simulation of different "what if" scenarios. This means that in a virtual environment, the planner can check the feasibility of orders on a schedule, create a Gantt chart, check for bottlenecks, order parts to assemble a product at the right time, and more. In this way, the assembly process is better organized, time losses are reduced, and the supply of products is better determined.

### 4.6. Discussion

According to the literature and the results, the presented approach is very useful and suitable for different types and sizes of manufacturing systems. The approach is especially useful for discrete manufacturing systems, which sometimes impose limitations. In our case, the results were based on a linear assembly line with different building blocks.

In the first step of the approach, the manufacturing building blocks were classified into four groups. The results show that each building block of the manufacturing system can be classified into one group. Unlike the paper [22], where the building blocks of the manufacturing system were divided into three groups, we used four groups. This division

proved to be very useful, as we described each building block with its own parameters that we need in the following steps.

In the second step, we defined all the processes in the manufacturing system. The result of the second step is a sequence of processes that follow from the first process (start of assembly) to the last process (the product is stored in the warehouse). We have developed a functional diagram that includes only the most important information about the processes.

The result of the third step is the definition of the parameters for each building block. Based on the parameters and already established methods for planning digital models, a digital model of the manufacturing system was created. The focus of this step is on defining the appropriate parameters rather than developing the digital model. The detailed procedure of developing a digital model is presented in other papers [46,47]. Using the aforementioned papers and [16,24], we found that there is a lack of approaches to defining parameters. The papers talk about data management, data transfer, data storage in general, but do not define exactly which parameters are required for a comprehensive description of the manufacturing system. The results of this step clearly show which parameters are needed to develop a digital model of the conveyor belt. By defining the parameters in this way, we can describe different sizes, types and shapes of the conveyor belt, regardless of the simulation software tool used.

The result of the fourth step is the connection of the digital model to the real manufacturing system, thus establishing a digital twin with automatic data transmission and two-way communication. Although several attempts to establish a digital twin [30–34] can be found in the literature, the presented literature does not give a clear answer on what data needs to be exchanged and in what form. In our approach, the digital twin obtains data from a real system with inductive sensors, a camera and RFID tracking technology. The example shows the setup and operation of a digital twin. Various digital agents play an important role in the operation of the digital twin.

The fifth step focuses on the visualization of the real manufacturing system. Nowadays, visualization is very important for different participants, from operators to management. The presented visualization shows a combination of results from a real manufacturing system enriched with data from the digital twin. A significant advantage is that the digital twin is the same for all participants, only the interpretation of the results is different. There are many studies in the literature that focus on process visualization [26,28], but there is a lack of comprehensive representations of visualization for different participants. This resonates with our reasons we developed the visualization the way we did.

## 5. Conclusions

There are a variety of different software solutions (e.g., Siemens Tecnomatix Plant Simulation) for developing digital models that are used to improve real manufacturing systems and processes. The improvements are mainly related to manufacturing process planning, order scheduling and also to simplification of the visualization for various participants. To obtain a digital twin, it is necessary to capture process parameters and synchronize the real system with the digital model.

The paper presents a five-step approach to planning the data-driven digital twin for discrete manufacturing systems: (i) Definition of manufacturing system. The approach is based on breaking down the manufacturing system into building blocks belonging into four groups: "Fabrication", "Logistics", "Storage", and "Inspection". (ii) Determining the sequence of processes in the manufacturing system. Classification from (i) forms the basis of identifying relevant parameters for planning the digital model (iii). The proposed approach shows which parameters are required for each group. In addition to the generalized data of the groups, each building block has its own specific data and specifications that need to be considered when planning a digital model. (iv) The digital model is upgraded to a digital twin with a feedback control loop to the real manufacturing system. The real manufacturing system sends initial data (the number of parts in buffers, WBPFP and WRG) to the digital model. Then the digital model sends control information back to the real

system. The digital twin is able to generate a suboptimal manufacturing plan, check the current state of the real manufacturing system before the process starts, correct the real system/process according to the simulation of the digital twin, etc. (v) A visualization is introduced for different participants. By using a digital twin, they can make decisions based on the parameters they need (data-driven decisions).

The approach is validated in a case study of Demo Center Smart Factory. It contains at least one example for each group of building blocks and presents the assembly of different products. Based on the case study, each step of the presented approach is clearly described and allows for the step-by-step guidance of the user. The digital model (for logistics) in our case study was created using the Siemens Tecnomatix Plant Simulation software tool, but the approach is useful for any simulation software tool.

Preparing the traditional manufacturing systems and processes, including the software equipment and visualization for digitization is still a significant challenge for the companies. In our experience, companies are struggling to define the right parameters that should be collected from the real systems and processes, ready as inputs to the digital models (digital twins) capable of performing real-time data analysis, process simulation and feedback control.

Practitioners can use the results of our work as a guideline for defining and preparing the right parameters for different manufacturing systems and processes. Each system or process has its own specific characteristics and therefore requires different essential parameters. The presented approach guides practitioners through five crucial steps that contain the important information about which parameters are necessary for the digital twins and consequently where to install the sensors to collect this data from the real system. Knowing this is important to avoid later modifications of the systems, which are sometimes difficult to accomplish.

Finally, our study helps experts and researchers to design modern manufacturing systems and processes that are ready to be upgraded with the digital twin. By using our proposed five-step approach, the experts and researchers are able to develop the specific data-driven digital twins that cover other systems and processes, which will be useful for the practitioners.

In our case, the presented approach has proved to be very useful for the industry, as practitioners follow the approach to prepare the manufacturing system data and processes and develop the digital twins.

We see future research in the development and improvements of the approach to develop a digital twin. We will focus on further development of (i) real-time communication between the digital model and the real manufacturing system, (ii) data analysis, and (iii) more person-oriented visualization of the output parameters of the digital twin. In addition, we would like to fully automate the processes contained in the steps, such as automatic inclusion of the layout of the manufacturing system, automatic definition of a building block's data, etc.

**Author Contributions:** Conceptualization, M.R. and N.H.; methodology, M.R.; software, M.R. and J.P.; validation, M.R., J.P. and M.S.; formal analysis, M.R. and J.P.; investigation, M.R.; resources, M.R., J.P., M.S. and N.H.; data curation, M.R. and J.P.; writing—original draft preparation, M.R. and J.P.; writing—review and editing, J.P., M.S. and N.H.; visualization, M.R. and J.P.; supervision, M.S. and N.H. All authors have read and agreed to the published version of the manuscript.

**Funding:** The work was carried out in the framework of the GOSTOP programme (OP20.00361), which is partially financed by the Republic of Slovenia, Ministry of Education, Science and Sport Republic of Slovenia (Ministrstvo za izobraževanje, znanost in šport) and European Union, European Regional Development Fund.

**Institutional Review Board Statement:** Not applicable.

**Informed Consent Statement:** Not applicable.

**Conflicts of Interest:** The authors declare no conflict of interest.

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
