# Peer review of "A Five-Step Approach to Planning Data-Driven Digital Twins for Discrete Manufacturing Systems"

_applsci, doi:10.3390/app11083639_

Round 1

Reviewer 1 Report

Thank you very much for giving me the opportunity to read this interesting article. It is a case study that illustrates the development of the digital twin for discrete manufacturing systems.

The study is composed of  4 sections: introduction, material and methods, case study, result and discussion, and conclusion.

The article has a good narrative flow with some typos. Try to avoid expressions like “so only” or similar.

Here you find the major comments to the article:

- In the introduction, please clarify the contribution of the study to the literature.

-  In the material and methods when you present the value of smart manufacturing (related to competitiveness) you should also the importance of its sustainable impacts (reduction of raw materials and in some case better work conditions). Here an example:

 Margherita, E.G., Braccini, A.M.: Industry 4.0 technologies in flexible manufacturing for sustainable organizational value: reflections from a multiple case study of italian manufacturers. Inf. Syst. Front. (2020).

- Add some reference in the materials part

- results and discussion: Please discuss the results with the literature pinpointing the contribution of the study

- Conclusion: You have to add some lines regarding the study implications for researchers and practitioners.

Here you find minor issues:

- Update the reference section.

- Put figure and table caption above or below the element according to the journal rule.

Author Response

Dear Reviewer,

We appreciate your contribution by giving us very useful suggestions and comments. With Your help, we are able to improve the article which we hope will be more understandable to a wider readership. We have gone through all of them carefully and taken them into account in the article content. All changes and all answers to your questions and comments are written in red text. Please find our itemized responses below and our revisions/corrections in the re-submitted paper.

Thanks again!

Reviewer 2 Report

The article presents in an accessible way the current and important from the production point of view issues related to creating digital twins for production systems. It precisely shows the technology of creating such systems. Describes important parameters of both real and virtual systems. It shows how to build such systems, choose parameters and how to synchronize the real and virtual model. The example presented in the work perfectly illustrates the issues discussed. Noteworthy is a very extensive list of references that may be helpful for anyone interested in the topic in question.

The composition of the article does not raise any objections.

Important and interesting article. I recommend.

Author Response

Dear Reviewer,

We are very glad that you evaluated the article as interesting and recommended for publication in proposed journal. We appreciate your opinion.

Thanks again!

Reviewer 3 Report

This paper focuses on a very important topic for the implementation of digital twins. The proposed strategy for planning of digital twins of manufacturing systems could be very useful in practical applications. The paper is well structured with solid theoretical explanations and experiment validations. 

I just have a few suggestions for possible improvement.

1) It would be better if Figure 2 can be replaced with a more detailed application architecture or reference architecture for implementing the proposed approach. Multiple function blocks and enabling technologies are mentioned in the paper. Is it possible to integrate them into one framework?

2) The data management issue seems not well explained. In section 4.5, it is mentioned "During each process, the values of the process parameters are collected and stored. The values are forwarded to the SCADA system...". How do you manage these data, i.e. data communication, storage, protocols/standards etc.?

Author Response

Dear Reviewer,

We appreciate your contribution by giving us very useful suggestions and comments. With Your help we are able to improve the article, which we hope will be more understandable to a wider readership. We have gone through all of them carefully and taken them into account in the article content. All changes and all answers to your questions and comments are written in blue text. Please find our itemized responses below and our revisions/corrections in the re-submitted paper.

Thanks again!

Round 2

Reviewer 1 Report

The authors addressed all my concerns.